# Memory-guided microsaccades

Konstantin F. Willeke [1,2,3,4], Xiaoguang Tian[1,2,3,4], Antimo Buonocore [1,2,4], Joachim Bellet[1,2,3],
Araceli Ramirez-Cardenas [1,3] & Ziad M. Hafed [1,2]

Despite strong evidence to the contrary in the literature, microsaccades are overwhelmingly described as involuntary eye movements. Here we show in both human subjects and monkeys that individual microsaccades of any direction can easily be triggered: (1) on demand, based on an arbitrary instruction, (2) without any special training, (3) without visual guidance by a stimulus, and (4) in a spatially and temporally accurate manner. Subjects voluntarily generated instructed "memory-guided" microsaccades readily, and similarly to how they made normal visually-guided ones. In two monkeys, we also observed midbrain superior colliculus neurons that exhibit movement-related activity bursts exclusively for memory-guided microsaccades, but not for similarly-sized visually-guided movements. Our results demonstrate behavioral and neural evidence for voluntary control over individual micro-saccades, supporting recently discovered functional contributions of individual microsaccade generation to visual performance alterations and covert visual selection, as well as observations that microsaccades optimize eye position during high acuity visually-guided behavior.

---

[1] Werner Reichardt Centre for Integrative Neuroscience, Tuebingen University, 72076 Tuebingen, Germany. [2] Hertie Institute for Clinical Brain Research, Tuebingen University, 72076 Tuebingen, Germany. [3] Graduate School of Neural and Behavioural Sciences, International Max Planck Research School, Tuebingen University, 72074 Tuebingen, Germany. [4]These authors contributed equally: Konstantin F. Willeke, Xiaoguang Tian, Antimo Buonocore. Correspondence and requests for materials should be addressed to Z.M.H. (email: ziad.m.hafed@cin.uni-tuebingen.de)

A vast majority of currently published research on micro-saccades, as well as almost all public discourse on them, refers to these tiny fixational eye movements, which occur when we attempt to fix our gaze, as involuntary or spontaneous (Supplementary Fig. 1). However, a multitude of evidence from the literature actually points in the opposite direction. For example, likelihoods of microsaccade direction, amplitude, and frequency are systematically modulated under a variety of conditions[1–17], and microsaccade generation is causally affected by activity in the midbrain superior colliculus (SC)[18–21] and the cortical frontal eye fields (FEF)[22], both involved in voluntary eye movement control[23]. Microsaccades can also be suppressed either voluntarily[24,25] or when perceptually challenging discrimination stimuli are expected[7,8], and they precisely re-align gaze on the fixated stimulus[26–29], even when behavioral tasks require peripheral-stimulus monitoring[3–5]. Finally, two highly experienced human subjects were described as being able to generate voluntary horizontal or vertical saccades as small as microsaccades[30].

The persistence of microsaccade descriptions as being involuntary and spontaneous (Supplementary Fig. 1), despite the above evidence, stems from a severe lack of explicit demonstration that these eye movements, like larger saccades, can be generated by naïve subjects at will, based on abstract task-defined instructions, and without visual guidance. The strongest such evidence[30], based on a small number of expert observers and only with cardinal movement directions, has gone without direct follow-up for more than 40 years. However, such explicit demonstration is critically needed, particularly in the present day, given the resurgence of the field[4,6,20] (Supplementary Fig. 1), and given that the occurrence of any individual microsaccade is now known to strongly alter even peripheral visual performance[3–5,31–35], providing an almost-deterministic link between covert visual attentional effects and the execution of such tiny eye movements[3–5]. Here, we provide such demonstration; by adapting, in a novel way, a behavioral task exercising voluntary eye movement control to the realm of microsaccades, we show in both naïve human subjects and monkeys that tiny eye movements overwhelmingly thought to be involuntary (Supplementary Fig. 1) can be generated at will, based on arbitrary instruction, and without any visual guidance. We also show that these eye movements are governed by highly specific SC neural circuitry.

## Results

**Memory-guided microsaccades are generated at will.** We first asked three male rhesus macaque monkeys to perform a novel memory-guided microsaccade task, which was adapted from the known memory-guided saccade task[36,37] (Methods). The monkeys were initially trained to maintain fixation while a brief (58 ms) flash was presented at an eccentricity ≥5°, greater than an order of magnitude larger than microsaccade amplitudes (most frequently defined as < 1°, ref. [38], but often significantly smaller, ref. [18]). The monkeys had to remember flash location for 300–1100 ms, after which the fixation spot was removed. The monkeys used fixation spot disappearance as the abstract instruction to voluntarily execute a timely eye movement toward the remembered flash location, and the target only reappeared after the eye movement had ended (Methods); there was no specific instruction to re-fixate the reappearing target, but the monkeys often did so nonetheless. Our task therefore required voluntary eye movement generation based on an arbitrary instruction that we enforced by task design (fixation spot disappearance) and without any visual guidance[36,37]; corrective re-fixation saccades were visually guided.

After initial training with large saccades, we tested the monkeys on pseudorandomly chosen stimulus locations, including at small eccentricities typically associated with microsaccades. We specifically interleaved small and large eccentricities across trials within any given session, and we also sampled different visual field locations. We did so in order to avoid overtraining any specific locations or eccentricities. We were struck by the fact that the monkeys very easily generated memory-guided movements even for target eccentricities as low as 6 min arc (0.1°), and without any need for special training. They simply generalized the rule that they had learned for eccentricities ≥5°. Figure 1a demonstrates this by way of an example trial from monkey N.

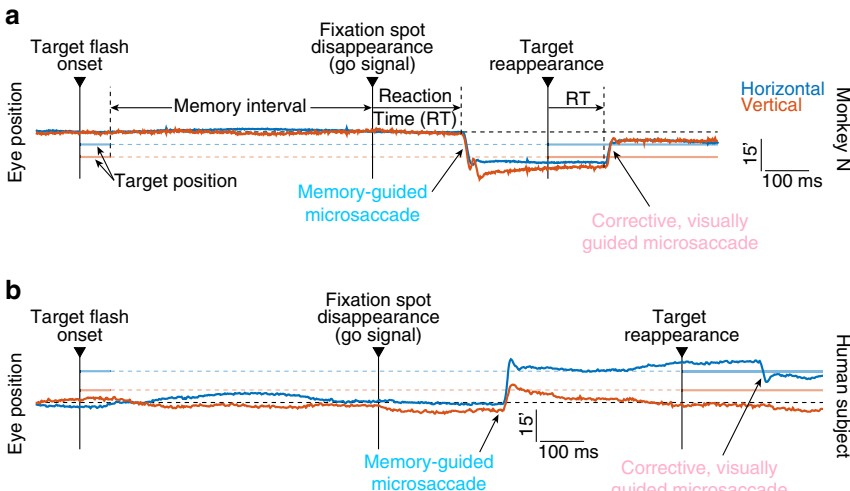

**Fig. 1** Memory-guided microsaccades in monkeys and humans. **a** Eye position measurements from an example trial in monkey N. Upward deflections in each trace mean rightward or upward eye position displacements, respectively, and the position scale bar denotes 15 min arc. A brief target flash appeared 6 min arc to the left of and 12 min arc below fixation. After a memory interval, the fixation spot disappeared, instructing a memory-guided eye movement towards the remembered flash location. The monkey generated a spatially accurate memory-guided microsaccade after a reaction time (RT) of approximately 206 ms from fixation spot disappearance. When the target reappeared, a smaller, corrective movement (which was now visually guided) was triggered after a shorter reaction time. **b** Similar observations from an example trial in a human subject. In this case, the target flash occurred at 18 min arc to the right of and 6 min arc above fixation. Note that the human data were recorded with a video-based eye tracker; monkey N data were obtained from a scleral search coil (Methods). Figures 2–4 and Supplementary Figs. 2–4 demonstrate the robustness of the observations in this figure across the population, and Figs. 6–7 and Supplementary Fig. 5 highlight underlying neural mechanisms

The monkey fixated steadily between flash onset (at a location 6 min arc left of and 12 min arc below fixation) and the instruction to trigger a movement (the go signal). The monkey then successfully generated an oblique downward–leftward microsaccade approximately 206 ms after the go signal (fixation spot disappearance). After target reappearance, the monkey corrected the remaining eye position error (due to overshoot in the memory-guided microsaccade) with a second visually guided microsaccade. Note how the reaction time of the corrective, visually guided microsaccade (from target reappearance) was comparable to that of the voluntarily generated memory-guided microsaccade (from fixation spot disappearance), and that the latter movement was directionally accurate (directed towards the remembered foveal flash location). Thus, spatially and temporally accurate microsaccades can be triggered at will, based on arbitrary instruction and without visual guidance (also see Supplementary Movies 1, 2).

Timely voluntary memory-guided microsaccade generation was consistent in all three monkeys, and also in seven human subjects (Methods; Fig. 1b shows an example trial from one human subject). For all flash eccentricities < 1° (the most typically cited amplitude threshold for microsaccades[18,38]; Methods), we plotted histograms of all microsaccade latencies after the go signal (fixation spot disappearance; Fig. 2a, d, g, j). This allowed us to evaluate the reaction times of instructed, memory-guided microsaccades. We also plotted the histograms of microsaccade latencies after target reappearance (Fig. 2m, p, s, v), for evaluating the reaction times of corrective, visually guided microsaccades. In all cases, memory-guided microsaccades had reaction times consistent with these movements being successfully triggered after task instruction[39,40]. Moreover, the latency distributions were similar to those of larger memory-guided saccades with flash locations >1° (Supplementary Fig. 2). Corrective, visually-guided microsaccades expectedly had shorter reaction times than memory-guided microsaccades (monkey N: 146 ms vs. 203 ms; monkey M: 160 ms vs. 212 ms; monkey P: 189 ms vs. 245 ms; humans: 189 ms vs. 252 ms; all $p < 10^{-50}$, t-test). Therefore, timely, voluntary microsaccades were generated at will after the abstract instruction to do so (fixation spot disappearance).

It may be argued that while memory-guided microsaccades did occur in a timely fashion, their likelihood of occurrence (what we defined as the success rate) was simply low and random as opposed to reflecting a deterministic, voluntary reaction to an explicit go instruction. After all, irrelevant abrupt onsets can reflexively reset microsaccade generation rhythms[41–43], suggesting that some aspects of microsaccade behavior may appear to be reflexive in nature. However, plotting cumulative probabilities of memory-guided microsaccades after the go signal (i.e. all trials with target eccentricity <1°), we found that almost every single instructed trial was associated with a response (Fig. 2b, e, h, k; erroneous trials were primarily ones with reaction times larger than our upper limit for acceptance, Methods). That is, using solely reaction time as the success criterion (Methods), the great majority of memory-guided microsaccade trials were successful. This was not the case for, say, microsaccades after flash onset at the beginnings of trials when monkeys were instructed to maintain fixation (Supplementary Fig. 3a–k). These early flash-induced microsaccades were not instructed, and therefore did not happen on every single trial (e.g. they were missing in the example trials of Fig. 1); their occurrence depended on a variety of other factors related to the quality of fixation control[3,5]. More importantly, success rate for the instructed memory-guided microsaccades (i.e. the fraction of trials in which a movement was successfully generated within a reasonable reaction time; Methods; Fig. 2b, e, h, k) was equal to or significantly higher than that for the corrective, visually guided microsaccades, which were

not instructed (Fig. 2n, q, t, w) (monkey N: $p = 0.028$; monkey M: $p = 0.5$; monkey P: $p < 10^{-8}$; humans: $p < 10^{-50}$, $\chi^2$-test). This indicates that task instruction was a strong impetus to generate the voluntary microsaccadic eye movements in our task. Expectedly, the lowest success rate for both memory- and visually guided (corrective) microsaccades was for the smallest target eccentricities (Fig. 2c, f, i, l, o, r, u, x).

Overall, these results indicate that memory-guided microsaccades can indeed be generated explicitly following abstract instruction (i.e. one that is learned from task design), and even more successfully than uninstructed corrective, visually-guided movements. These are hallmarks of voluntary behavior. Interestingly, when visually-guided microsaccades were explicitly instructed in a separate condition (the delayed, visually guided saccade task; Methods), their success rates were also less than those of the memory-guided microsaccades (Supplementary Fig. 4a, d). This is largely due to the fact that the monkeys reflexively fixated the visually persistent target in this task paradigm even before the go signal.

**Memory-guided microsaccades are spatially accurate.** High success was also evident in the directional accuracy of memory-guided microsaccades and not just in their reaction times (Fig. 2). For the same data in Fig. 2, we now plotted the angular directional difference between memory-guided microsaccades and target locations (Fig. 3b, f, j, n). That is, we plotted the direction error of the movements. We also plotted the same measure for the uninstructed corrective, visually guided microsaccades (Fig. 3c, g, k, o); in this case, we compared corrective microsaccade direction to the direction of the reappearing target relative to gaze position after the memory-guided microsaccade had ended (Methods). Once again, in both monkeys and humans, memory-guided microsaccades were significantly more directionally accurate than the uninstructed corrective, visually guided movements. We confirmed this statistically by comparing the distribution of absolute directional errors for memory-guided microsaccades to that of absolute directional errors for corrective, visually guided microsaccades. In monkey N, the former was 20.4° ± 5.9° 95% c. i. and the latter was 27.1° ± 1.3° 95% c. i. ($p < 10^{-4}$, Watson-Williams test, circular analog of the two-sampled $t$-test); for monkey M, the results were 23.4° ± 2.5° 95% c. i. and 31.5° ± 1.1° 95% c. i. ($p < 10^{-10}$); and for monkey P, they were 33.8° ± 2.4° 95% c. i. and 51.2° ± 1.4° 95% c. i. ($p < 10^{-50}$). Similarly, for humans, the directional errors were 23.8° ± 0.8° 95% c. i. versus 43.1° ± 0.9° 95% c. i. ($p < 10^{-50}$). When instructed visually guided microsaccades were tested separately (during the delayed, visually guided saccade task; Methods), directional accuracy was restored to the levels of memory-guided microsaccades (Supplementary Fig. 4b, e), consistent with prior results[26]. Therefore, we observed high success in both reaction time (Fig. 2) and directional accuracy (Fig. 3) of instructed memory-guided microsaccades.

It may also be argued that the directional accuracy described above was not necessarily a result of a voluntary effort of will to follow task instruction, but instead an outcome of covert attentional shifts. For example, monkey[7] and human[1,2] microsaccades exhibit direction biases in covert attentional tasks, and it could be the case that the requirement to remember target location was enough to bias microsaccade directions near the ends of trials (perhaps due to sustained covert attention). However, if so, such a bias should have also appeared during the memory interval itself. We therefore analyzed microsaccade directions in the final 250 ms of the memory interval, during instructed fixation (Fig. 3a, e, i, m). Directional accuracy, relative to target location, was much worse than in the instructed

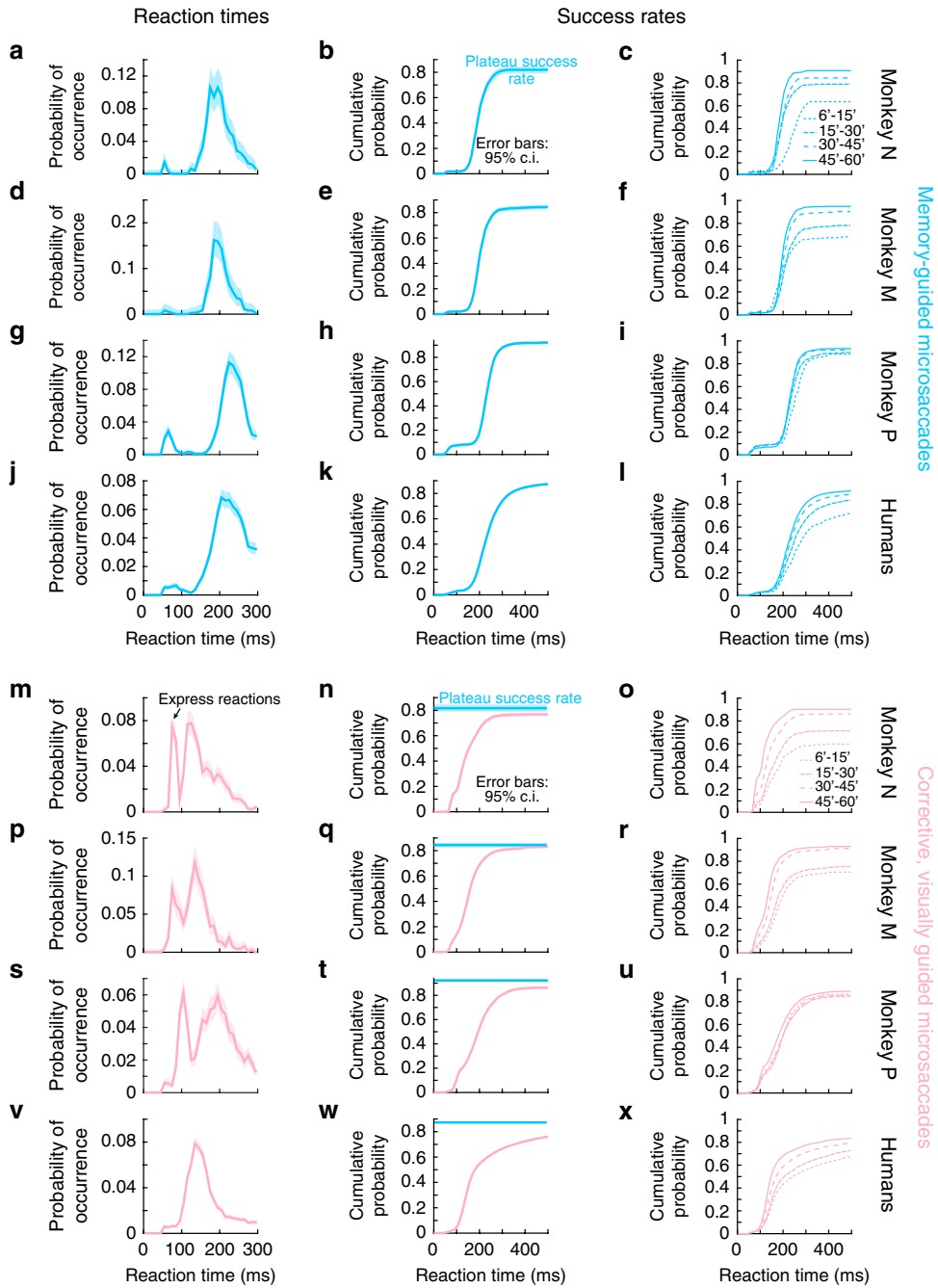

**Fig. 2** Memory-guided microsaccades had reaction times and success rates consistent with being genuine responses to task instruction. **a** Reaction time distribution for memory-guided microsaccades in monkey N. The data shows all trials with target eccentricities <1º (Methods). The monkey had a reaction time distribution typical of saccadic responses. Error bars denote 95% confidence intervals, and the histogram was normalized by the total number of trials. **b** Cumulative probability of the same data, demonstrating a plateau of >80% success rate; the monkey successfully generated a memory-guided microsaccade (that is, within a reasonable reaction time; Methods) on the great majority of trials. Error bars denote 95% confidence intervals. **c** Same data as in **b** but separated by target eccentricity (inset legend). Even the smallest target eccentricities were associated with a majority of successful reactions. **d–f** Similar observations from monkey M. **g–i** Similar observations from monkey P. **j–l** Similar observations from our human subjects (Methods). **m–x** We repeated the same analyses above, but now for corrective, visually guided microsaccades occurring after target reappearance. Reaction times (**m**, **p**, **s**, **v**) were faster than for memory-guided microsaccades, and often exhibited express reactions (e.g. arrow in **m**). However, success rates were significantly lower than in the instructed memory-guided microsaccades (**n**, **q**, **t**, **w**; the horizontal blue line in each panel, and associated 95% confidence interval, replicates the plateau success rate from the corresponding panel in **b**, **e**, **h**, **k**). Success rates of corrective, visually-guided microsaccades also depended on target eccentricity (**o**, **r**, **u**, **x**) similarly to memory-guided microsaccades. $n = 781, 1,602, 2,235, 7,402$ memory-guided trials in monkey N, monkey M, monkey P, and humans, respectively. $n = 2774, 4471, 3461, 10,866$ visually guided trials in monkey N, monkey M, monkey P, and humans, respectively

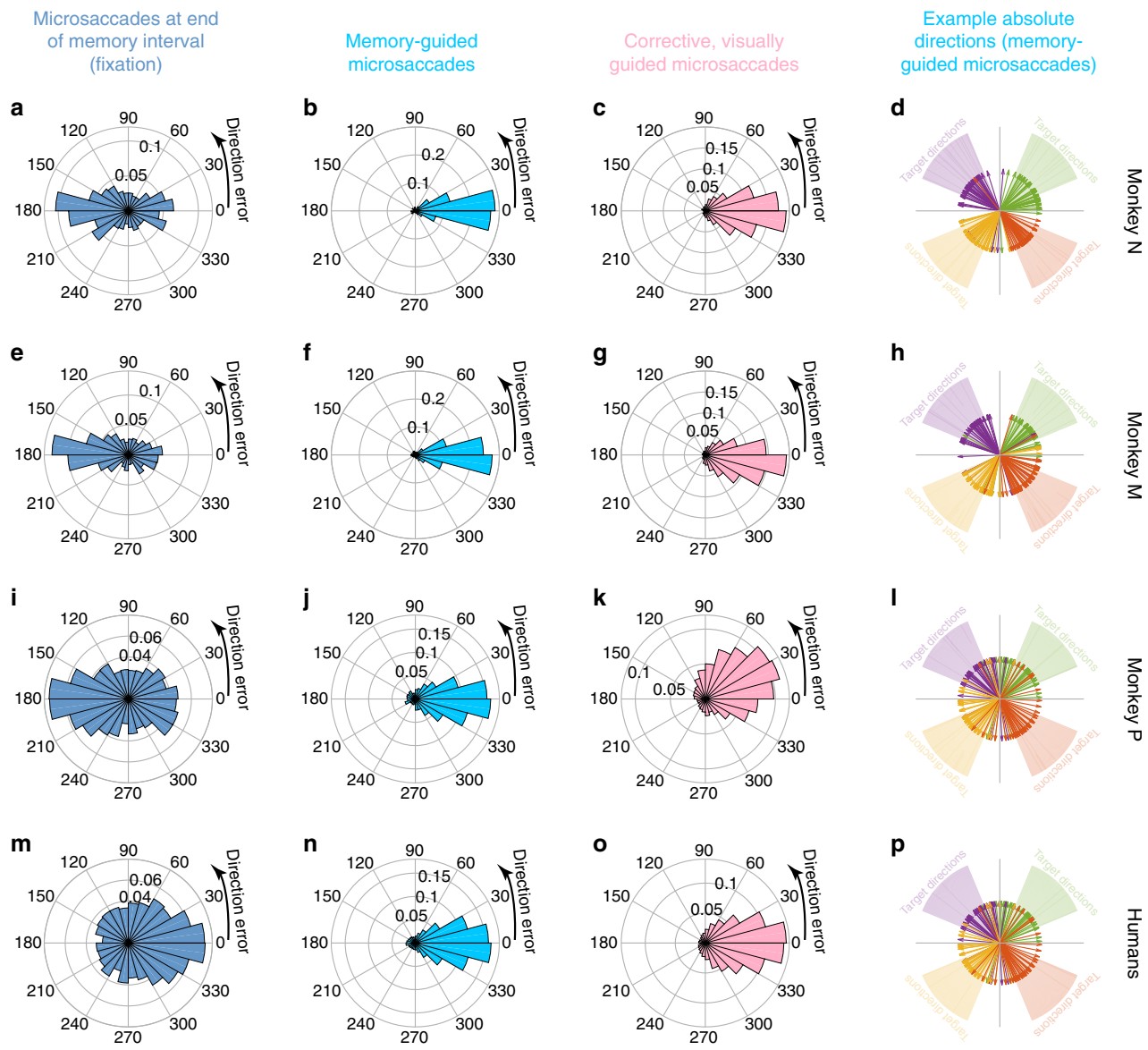

**Fig. 3** Memory-guided microsaccades had better directional accuracy than either microsaccades during fixation or corrective, visually-guided microsaccades. **a** We plotted the angular distribution of microsaccade direction error (that is, the difference between target direction and microsaccade direction; Methods). In this measure, if a microsaccade has the same direction as the target direction, then the movement has zero direction error. In this panel, this measure is shown for all microsaccades from monkey N occurring in the final 250 ms of fixation before fixation spot disappearance. Microsaccades could occur either towards or away from the remembered target location. The histogram was normalized by the total number of observations. **b** The same monkey exhibited highly accurate microsaccade directions toward the remembered target location when explicitly instructed to do so. **c** After target reappearance, corrective, visually-guided microsaccades were also target-directed, but their direction errors were significantly more variable than the instructed movements in **b** (see statistics in text). Thus, instructed memory-guided microsaccades were highly directionally accurate. **d** Each group of colored arrows shows the directions of 40 randomly selected memory-guided microsaccades when remembered target directions were drawn from the underlying corresponding-colored faint cone. Directionally accurate microsaccades could be generated for all oblique target directions. **e**–**h** Similar observations for monkey M. **i**–**l** Similar observations for monkey P. **m**–**p** Similar observations for our human subjects. In all cases, directional accuracy was the highest for the instructed, memory-guided movements and not for fixational microsaccades (**a**, **e**, **i**, **m**) or corrective, visually guided ones (**c**, **g**, **k**, **o**); also see Supplementary Fig. 3 for flash-induced microsaccades. For the leftmost column, $n = 1,432$, 2,875, 4,870, 2,831 microsaccades for monkey N, monkey M, monkey P, and the human subjects, respectively. The numbers of trials for memory-guided or corrective, visually guided microsaccades are the same as in Fig. 2

memory-guided microsaccades (Fig. 3b, f, j, n) ($p < 10^{-50}$ for comparing absolute directional error in memory-guided microsaccades and microsaccades at the end of the memory interval in each of the monkeys and also in the human data; Watson-Williams test). In fact, the fixational microsaccades (at the end of the memory interval) were biased either away (monkeys) or towards (humans) the target location. This difference in directional bias between species (for the same trial-duration distributions) likely reflects expected differences in the temporal dynamics of microsaccadic oscillations between humans and monkeys[3,33,42].

We also analyzed microsaccades immediately after target flash onset, to compare potential reflexive stimulus-induced effects (perhaps due to exogenous covert attentional effects after target

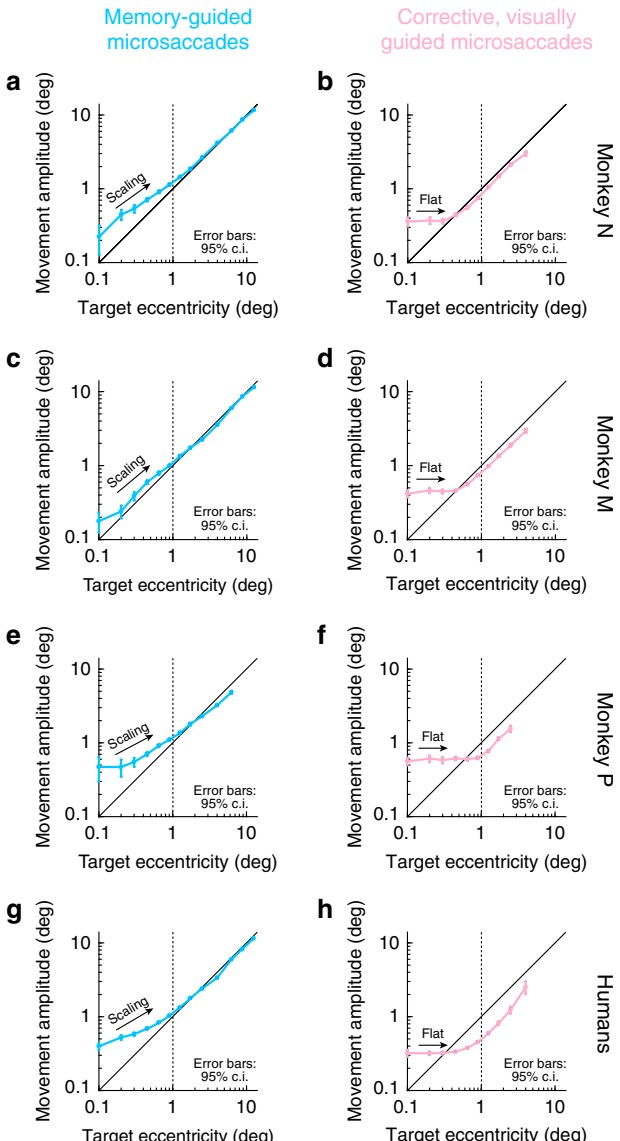

**Memory-guided microsaccades**

**Corrective, visually guided microsaccades**

Fig. 4 Memory-guided microsaccade amplitudes scaled better with target eccentricity than corrective, visually-guided microsaccades. **a** For all target eccentricities in monkey N, we plotted memory-guided movement amplitude as a function of target eccentricity. The data below 1º target eccentricity (vertical dashed line) are from the same trials described in Figs. 1–3 for reaction times, success rates, and directional accuracy. Here, we show their amplitude scaling. Memory-guided microsaccades increased in amplitude with increasing target eccentricity, consistent with them being genuine responses to task instruction, but they showed systematic overshoot (e.g. Figure 1). The overshoot disappeared for larger eye movements (also see Fig. 5 for a potential explanation). Target eccentricities were binned at the following non-overlapping bin-center values: 0.05, 0.15, 0.25, 0.35, 0.55, 0.75, 1.05, 1.45, 2, 3, 5, 7.5, 10, 15; error bars denote 95% confidence intervals. **b** For similar small target eccentricities, corrective, visually-guided microsaccades overshot the target more than the instructed memory-guided microsaccades, and their amplitudes did not scale with target eccentricity as well as the memory-guided movements (despite the presence of a visual target). Larger corrective movements were more accurate. **c**, **d** Similar observations in monkey M. **e**, **f** Similar observations in monkey P. **g**, **h** Similar observations in the human subjects. In all cases, despite the amplitude overshoot for the smallest memory-guided microsaccades (see Fig. 5), instructed memory-guided microsaccades exhibited better amplitude scaling with increasing target eccentricity than corrective, visually-guided microsaccades (also see Fig. 5). $n = 2{,}947$, 4,931, 3,972, 10,890 for memory-guided movements of all amplitudes in monkey N, monkey M, monkey P, and the human subjects; $n = 2{,}528$, 4,086, 2,674, 7,002 for corrective, visually-guided movements of all amplitudes in monkey N, monkey M, monkey P, and the human subjects

flash onset) to instructed movements after fixation spot disappearance (Supplementary Fig. 3c, f, i, l): other than an expected minority of express visually driven reactions immediately after flash onset[5], the directions of flash-induced microsaccades were much more variable than the directions of instructed memory-guided microsaccades. Therefore, even transient, exogenous covert attention shifts[1,2] are not sufficient to explain the directional accuracy of the instructed memory-guided microsaccades (Fig. 3b, f, j, n).

It is also important to note that voluntary, memory-guided microsaccades could be triggered in all directions. Thus, directional accuracy was not restricted to, say, only horizontal or only vertical directions, one limitation in the original Haddad and Steinman study exploring voluntary triggering of small eye movements[30]. To illustrate this, Fig. 3d shows raw plots of example memory-guided microsaccade direction vectors, color-coded by target locations in each of the four quadrants from monkey N. The other two monkeys and the human subjects also generated directionally accurate memory-guided microsaccades in all directions (Fig. 3h, l, p). Thus, not only did all subjects and monkeys successfully generate untrained voluntary eye movements for small-eccentricity target locations, but these eye movements were also highly directionally accurate. The

directional accuracy was, once again, better than in the uninstructed corrective, visually-guided movements (Fig. 3c, g, k, o).

We next investigated memory-guided microsaccade amplitudes. We found that such amplitudes also scaled with remembered target eccentricity, consistent with the notion that there was voluntary generation of spatially accurate (in both direction and amplitude) microsaccadic eye movements in the task. For example, in Fig. 4a, c, e, g, we plotted the amplitude of memory-guided microsaccades as a function of remembered target eccentricity (this figure also included larger saccades and eccentricities for comparison). We used logarithmic scaling to magnify the small eccentricities associated with microsaccades[19]. There was a small overshoot in movement amplitude for the smallest target eccentricities, but the amplitude of eye movements systematically increased with increasing eccentricity of the remembered target location. That is, despite the initial overshoot, there was scaling of microsaccade amplitudes with remembered target eccentricity. This scaling of memory-guided microsaccade amplitude was true in all 3 monkeys (Fig. 4a, c, e), as well as in the human subjects (Fig. 4g). Interestingly, when the monkeys or subjects made uninstructed corrective, visually-guided microsaccades in the same task, the amplitudes of these visually-guided movements (relative to target eccentricity) were worse than for the instructed memory-guided microsaccades (i.e. did not scale appropriately; Fig. 4b, d, f, h). Larger corrective movements were accurate in amplitude. Similarly, when visually-guided microsaccades were instructed (in the delayed, visually-guided saccade task; Methods), the amplitudes were more accurate (as expected[26]), and overshoot did not occur as strongly for the tiniest of eccentricities as in the case of the memory-guided microsaccades (Supplementary Fig. 4c, f). These results are consistent with all other results above, demonstrating willful generation of individual eye movements of microsaccadic sizes,

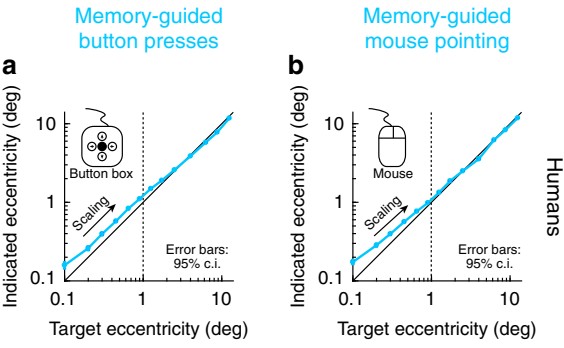

**Fig. 5** Overshoot was not restricted to microsaccades; perceptual localization of foveal remembered targets was also expanded in an illusory manner. **a** In humans, we tested perceptual localization of remembered target locations without explicitly requiring an eye movement response. In the button-press task, subjects used a button box to move a displayed cursor to the remembered target location (Methods). We analyzed their perceptual reports similarly to eye movement reports. The same overshoot for small target eccentricities was observed. Error bars denote 95% confidence intervals. **b** We also ran a variant of the task (the mouse pointer task; Methods), in which we also maintained a visual reference at fixation (the fixation spot). The same overshoot for foveal target eccentricities was still observed. Therefore, the overshoot of memory-guided microsaccades in Fig. 4 was likely a general property of representing foveal space in spatial working memory, as opposed to an intrinsic inability to generate small memory-guided microsaccades. $n = 6,405$ and $9,062$ trials for the button-press and mouse pointer tasks, respectively

based on abstract task-defined instruction to do so (fixation spot disappearance).

It is also interesting that the latency distributions for the corrective, visually-guided microsaccades in the same task (Fig. 2m, p, s, v) showed suggestive evidence of the oculomotor system being implicitly aware of residual errors after the previous memory-guided saccades or microsaccades (e.g. potential overshoot). Specifically, there was a distribution of express microsaccades with very short reaction times of <120 ms after target reappearance. Since express microsaccades are most likely when a movement plan is already well on its way at the time of stimulus onset[5], the occurrence of such express microsaccades in the corrective, visually-guided movements after target reappearance indicates that the previously-occurring memory-guided movements were indeed internally accounted for. This provides further evidence that memory-guided microsaccades are voluntarily triggered, and it is similar to larger saccades[44].

**Foveal working-memory expansion even with manual responses.** Despite the timeliness (Fig. 2a, d, g, j), success rate (Fig. 2b, e, h, k), and directional accuracy (Fig. 3b, f, j, n) of memory-guided microsaccades, their amplitude overshoots for some eccentricities (Fig. 4a, c, e, g) may be interpreted as a failure to make genuine target-directed voluntary eye movements of small amplitude. However, an alternative possibility is that the overshoots were instead a property of the encoding of foveal locations in spatial working memory, and not an execution limitation by the oculomotor system. For example, this could be a result of foveal magnification of neural tissue, which also exists in the SC to a much larger extent than previously thought[45]. To investigate this alternative, we ran our human subjects on two control tasks not explicitly requiring an eye movement to report the perceived location of the remembered target flash (Methods). Rather, we asked the subjects to perceptually report the remembered target location, and they did so via button or computer-mouse presses (Methods). In one variant, the button-press task, subjects were

free to move their eyes while responding, and they had up to 20 s to finalize their response; in another variant, the mouse pointer task, subjects pointed to the remembered location (with mouse cursor) while still maintaining fixation. In both cases, amplitude overshoot for the smallest target eccentricities was still clearly evident (Fig. 5), much like when eye movements were used to perform the task (the memory-guided microsaccade task). We should also note here that directional accuracy was also very high in the memory-guided manual tasks, similar to Fig. 3. Therefore, memory-guided microsaccade amplitude overshoot (Fig. 4a, c, e, g) was not a failure of the oculomotor system to voluntarily generate small eye movements. Instead, it reflected an intriguing foveal expansion in the representation of space during working memory (Fig. 5). Consistent with this, and as stated above, microsaccade amplitude overshoot was less evident with instructed delayed, visually-guided microsaccades (Supplementary Fig. 4c, f).

**Superior colliculus bursts for memory-guided microsaccades.** The strongest evidence that we obtained for voluntary control over microsaccades emerged when we recorded SC neural activity. In monkeys N and M, we recorded visual, visual-motor, and motor (saccade-related) SC neurons (Methods) during the memory-guided saccade task. We specifically aimed to sample sites in the rostral SC, in which microsaccade-related activity is expected[18,19,21], but we also sampled slightly more caudally (i.e. more eccentrically) for completeness (Methods). We inserted multi-electrode linear arrays into the SC, and we used offline sorting to isolate individual putative single units (Methods). Our first goal was to ask whether SC neurons responded at all during memory-guided microsaccades. For example, some SC neurons do not emit saccade-related responses in the absence of a visual target[46–48], and it is not clear whether this can happen for microsaccades or not.

Individual SC neurons did indeed emit saccade-related motor bursts for memory-guided microsaccades. For example, in Fig. 6a, we plotted neural activity from a sample neuron from monkey M. We identified a set of memory-guided microsaccades of amplitude <1º and direction toward the neuron's movement response field (RF), shown in Fig. 6b (neural RF's were assessed as described in Methods, and the movements included in Fig. 6a were directed towards the RF hotspot location and with angular direction within ±90º from the vector connecting gaze center to this location). We plotted radial eye position aligned on microsaccade onset and also firing rate (individual rows of tick marks indicate action potential times, with each row representing an individual movement; this formatting is similar to that in Fig. 1c of ref. [18]). Memory-guided microsaccade-related SC discharge was clearly present even for memory-guided movements as small as 6 min arc, and similar to expected saccade-related discharge in general. Therefore, individual voluntary, memory-guided microsaccades are associated with SC neural discharge.

Not only did we find microsaccade-related responses for memory-guided microsaccades (Fig. 6), but there were also neurons that discharged exclusively for memory-guided microsaccades and not for similarly sized corrective, visually-guided microsaccades. Figure 7a shows an additional example neuron from the same monkey as in Fig. 6, now demonstrating this unexpected phenomenon. We plotted an estimate of the neuron's movement field on log-polar coordinates up to 1º eccentricity. In the left panel, we plotted the movement field of the neuron when memory-guided microsaccades were generated. Clear discharge for specific movement vectors was evident. In the right panel, we plotted the movement field of the same neuron when corrective,

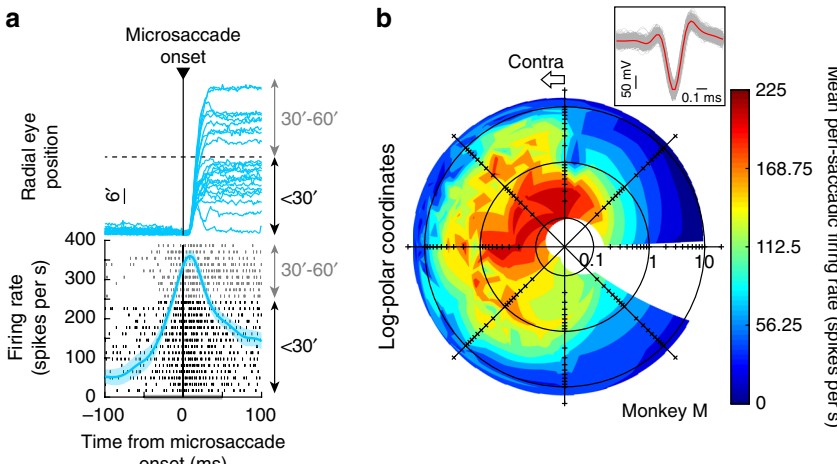

**Fig. 6** Superior colliculus (SC) activity during memory-guided microsaccades. **a** The top panel shows radial eye position aligned on microsaccade onset (for memory-guided microsaccades <1°). All traces were aligned to start (on the vertical axis) from the same eye position. Most movements had a radial amplitude <30 min arc (0.5°). These movements were selected relative to the response field (RF) as assessed from the analysis in **b**. The bottom panel shows microsaccade-aligned firing rate. Classic SC microsaccade-related activity[18] was observed for memory-guided microsaccades. Each row of tick marks shows the neuron's discharge for an individual movement from the top panel, and the rows are sorted by memory-guided microsaccade amplitude (black for <30 min arc; gray for 30–60 min arc). Even the smallest memory-guided microsaccades were associated with SC microsaccade-related discharge[18]. The gray bar on the x-axis indicates the analysis window used in **b**. Error bars denote 95% confidence intervals. **b** Movement RF of the same neuron. For each movement displacement, plotted in log-polar coordinates[19], we plotted mean firing rate from the measurement interval in **a**. The neuron, from the right SC, exhibited a movement RF on the contralateral side along an oblique upward direction. The neuron clearly responded for memory-guided microsaccades as small as 0.1° (6 min arc). The origin of the log-polar plot denotes 0.03° radial amplitude. The inset shows spike waveforms from every 100th spike recorded in the task (gray; 682 waveforms); red shows the average spike waveform (and 95% confidence intervals) across all recorded spikes

visually-guided microsaccades of similar amplitudes and directions were generated. The neuron now did not emit any movement-related discharge at all (compare to the left panel), even though movements of similar sizes were executed. These results were also obvious when we plotted firing rates as a function of time from microsaccade onset in the two conditions (Fig. 7b). Activity was present when memory-guided microsaccades were executed, but not when visually-guided movements of the same amplitude and direction range were generated. Activity for these latter movements looked indistinguishable from activity during baseline fixation before trial onset (Fig. 7b, black curve; Methods define baseline intervals). Therefore, not only was there strong behavioral evidence that naïve monkeys and humans can generate voluntary, memory-guided microsaccades (Figs. 1–4), but there were also individual SC neurons showing exclusive responses only for memory-guided microsaccades but not for similarly sized visually guided ones.

We found 41 SC neurons exhibiting exclusive responses for memory-guided microsaccades; these constituted 22% of SC neurons with saccade-related discharge in either memory or visual condition (Fig. 7e, f). Our criteria for including neurons in this accounting was the observation of visual and/or movement discharge for eccentricities <1° (Methods). We referred to these neurons as ER-Neurons, for "Exclusively Responding" (Methods). Their average firing rate modulations for foveal visual target onsets and for microsaccades are shown in Fig. 7g, h (rightmost column), and these modulations are consistent with those in the example neuron of Fig. 7a, b. Interestingly, for larger saccades, ER-Neurons did exhibit saccade-related activity for visually guided eye movements (Supplementary Fig. 5), suggesting that memory-guided microsaccades may have recruited additional neural resources from SC saccade-related neurons that would normally be recruited for slightly larger eye

movements when visual targets are present (Supplementary Fig. 5). Consistent with this, RF hotspots for ER-Neurons tended to be more eccentric than other neuron types in our database (Fig. 7f).

Interestingly, we also found all other classic saccade-related SC responses to be present as well for memory-guided microsaccades. For example, we found visual-only neurons responding to foveal flash onset in the beginning of a trial but not during the actual memory-guided (or visually guided) eye movements (Fig. 7g, h, leftmost column). We also found visual-motor neurons, exhibiting both a visual response and either a memory- or visually-guided microsaccade response (Fig. 7g, h, second column). Motor neurons that responded for both memory- and visually guided microsaccades, but not to flash or target onset, were also present (Fig. 7g, h, third column). Finally, in addition to the ER-Neurons (e.g. Fig. 7a, b), we discovered neurons exhibiting the exact opposite profile: these were visually-dependent microsaccade-related neurons, extending earlier findings of SC visually-dependent saccade-related neurons (VDSR-Neurons[46–48]) to the realm of microsaccades. One example such neuron is shown in Fig. 7c, d. The neuron exhibited microsaccade-related activity exclusively during visually-guided microsaccades, but not during similarly sized memory-guided microsaccades. This neuron type (Fig. 7g, h, fourth column) was not described previously in refs. [18,19]. Overall, in the memory-guided microsaccade task, the different classes of neurons that we encountered (Fig. 7g, h) were distributed in our database (Methods) as shown in Fig. 7e.

Our results demonstrate that memory-guided microsaccades are generated at will: based on abstract task-defined instruction, without visual guidance, with high fidelity in spatial accuracy, and with underlying SC circuitry supporting all different aspects of memory- or visually-guided behavior. These results demonstrate

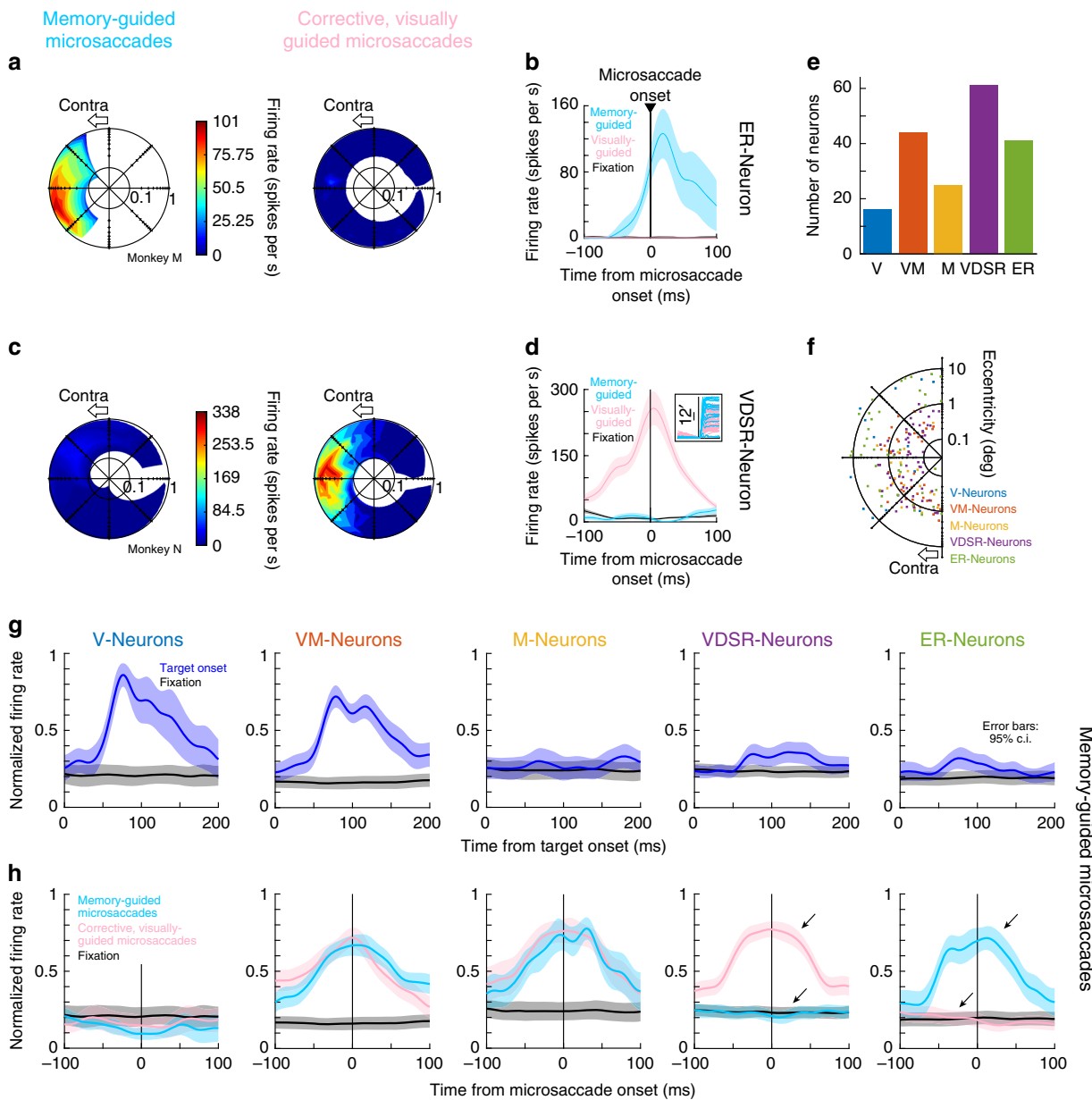

**Fig. 7** Double dissociation of SC microsaccade-related discharge for visually- and memory-guided microsaccades. **a** Movement RF's of an example SC neuron. Left shows peri-movement discharge (like in Fig. 6b) for memory-guided microsaccades; right shows the same for corrective, visually-guided microsaccades. The neuron was silent for the visually-guided movements, but not for similar-sized memory-guided ones. **b** Peri-movement discharge of the same neuron under the two conditions. The black curve (when visible) shows the neuron's activity during fixation before trial onset; the neuron only discharged for memory-guided, but not visually guided, microsaccades. Error bars denote 95% confidence intervals. **c, d** Same as **a, b** but for an example neuron only showing microsaccade-related discharge for visually guided movements (similar to visually dependent saccade-related, VDSR, neurons[46]). The inset shows (as an example) the analyzed eye movements (like Fig. 6a) in the two conditions, demonstrating clear overlap (similar overlap was present in **a, b**). **e** Distribution of the different neuron types (V = Visual; VM = Visual-motor; M = Motor; VDSR = Visually-dependent saccade-related; ER = Exclusively responding for memory-guided microsaccades). **f** Movement RF hotspots for the neurons. ER-Neurons tended to be more eccentric than other neurons exhibiting microsaccade-related discharge, consistent with a foveal expansion for memory-guided microsaccades (see text and Supplementary Fig. 5). **g, h** Peri-stimulus (**g**) or peri-microsaccadic (**h**) discharge for all of our encountered neuron types during microsaccade generation. Each neuron's response was first normalized before averaging responses across neurons. V- and VM-Neurons exhibited strong visual responses; VM- and M-Neurons showed movement-related activity regardless of stimulus; VDSR-Neurons only showed movement-related activity in the presence of a visual target; ER-Neurons only showed movement-related activity in the absence of a visual target. In all cases, we analyzed neural activity for microsaccades (<1º) toward the RF. Error bars denote 95% confidence intervals

that overwhelming descriptions of microsaccadic eye movements as being involuntary and spontaneous (Supplementary Fig. 1) are too simplistic and can mask the significance of these tiny eye movements in modulating perceptual and attentional performance.

## Discussion

We were motivated by a frustrating conundrum in the field of microsaccade research. Haddad and Steinman[30] showed in 1973 that two expert human observers were able to voluntarily trigger a microsaccade along one of the cardinal directions. Their study

has become virtually forgotten, especially after the resurgence of the field of microsaccade research from its almost-complete death at the end of the previous century[6] (Supplementary Fig. 1). We decided to revisit this particular study, but avoid some of its methodological constraints (such as the use of only two expert subjects, the limited amount of data, and the lack of neurophysiological insights), and we were able to demonstrate clear voluntary control over individual microsaccades even in naïve human subjects and monkeys. In doing so, we were also able to additionally demonstrate an ability for microsaccades to be driven by foveal visual working memory, in a novel adaptation of the memory-guided saccade task.

We were struck by how easily memory-guided microsaccades could be generated, with no need for special training. Monkeys simply generalized the task rule that they had learned for larger saccades, and humans were only verbally instructed to "look at the remembered target location". We believe that our results overturn an overwhelming discourse on microsaccades, even in public understanding of these eye movements, as being involuntary and spontaneous (Supplementary Fig. 1). Indeed, and as stated above, despite strong supportive evidence like that cited above and in Introduction, a surprisingly large fraction of references to microsaccades in the literature (especially in the last two decades) asserts that these eye movements are involuntary eye movements (Supplementary Fig. 1).

We believe that explicit demonstration of behavioral and neural evidence for voluntary control over individual microsaccades, in our novel adaptation of the memory-guided saccade paradigm, is absolutely necessary, especially in the present day when there is both a renewed interest in microsaccades as well as continued intense debates about their functional roles in influencing brain activity and behavior. Consider, for example, the field of covert visual attention. A long history of research has assumed for several decades that microsaccades either do not happen in these tasks at the time of task-relevant events, or that they happen in a random and spontaneous manner that should not influence the interpretation of results. However, it is now known that this view is too simplistic[1-5,31-34]. For this important revelation to become appreciated more widely, and in other fields related to visual and cognitive neuroscience, explicit demonstration of voluntary control over individual microsaccades, like we have shown in this study, is firmly needed.

The strongest evidence that we found for voluntary control over individual microsaccades is related to how the SC is involved in generating memory-guided versions of these eye movements. We were particularly intrigued by the ER-Neurons (Fig. 7). These neurons are SC neurons that discharge exclusively for memory-guided microsaccades but not for similarly sized visually-guided microsaccades. It would be highly interesting to investigate the mechanisms associated with these ER-Neurons in more detail in future studies. For example, the fact that these neurons discharge (i.e. exhibit saccade-related activity) for larger visually-guided saccades might suggest an expansion of RF's towards the fovea when spatial working memory is engaged (Supplementary Fig. 5). Prior work is inconclusive about how SC RF's change during spatial working memory tasks[47,49], and there is no systematic investigation, to our knowledge, of foveal spatial representation in working memory anyway. Therefore, this issue needs to be investigated in future follow-up studies. This can allow understanding the entire transformation from visual input (flash onset) to memory representation in the memory interval, and finally to movement generation. Relevant questions could include whether RF's change during the memory interval as well, or only at the time of memory-guided microsaccade generation (like in Fig. 7a, b, Supplementary Fig. 5). Importantly, studying this in the SC would clarify a sub-cortical involvement in working memory,

which is a topic that is generally investigated more strongly from a cortical perspective[36,50]. Studying this in the SC would also link SC activity to mechanisms of spatial updating of memory representations across eye movements[51,52].

We were equally intrigued by the VDSR-Neurons[46-48] that we found (Fig. 7c, d, g, h). In the past approximately four decades since they were first reported on in the primate SC, very few studies have mentioned these neurons, which only exhibit saccade-related discharge if saccades are directed towards a visual target. Moreover, these few mentions were only made from the perspective of larger saccades. Here, we found that VDSR-Neurons also exist in the rostral SC generating[18] microsaccades. That is, we found microsaccade-related neurons that only showed movement-related discharge if there was a visual target guiding microsaccade generation; the same neurons did not discharge when the microsaccades were generated toward a blank. We find these neurons interesting because they suggest that the oculomotor system, even for microsaccades, incorporates information about visual features at the time of saccade execution. It would be interesting to relate such incorporation of visual information in saccade commands to cortical interactions between saccades and visual representations[53] and to shape guidance of saccades[53] and microsaccades[42]. Because SC neurons exhibit visual-feature tuning in visual responses[54-56], it would also be interesting to explore how these VDSR-Neurons behave during saccades to different visual objects or features.

In all, our neural results with visual, visual-motor, and motor neurons all being involved in microsaccade generation confirm views that microsaccades are essentially indistinguishable from larger saccades in terms of neural mechanisms[20]. This is consistent with a variety of neurophysiological evidence in multiple areas[19,21,22,57,58], and it also gives further credence to the notion that these eye movements are indeed voluntary movements, despite the opposite description in a large portion of the literature (Supplementary Fig. 1). The visual responses, in particular, are worth further investigation, both for understanding SC foveal vision[45] in general, and also for exploring the speeded reactions to target reappearance in the corrective, visually-guided microsaccades (Fig. 2m, p, s, v).

Behaviorally, there was a remarkable difference between instructed and uninstructed movements in general. Most strikingly, the corrective, visually-guided microsaccades in the memory-guided task were both less successful (i.e. less often generated) and also sloppier in direction and amplitude (e.g. Figures 2–4) than the instructed memory-guided movements, even though these latter movements did not have explicit visual guidance. Only when we instructed visually guided saccades (in the delayed, visually guided saccade task) did visually guided microsaccades appear more spatially precise (Supplementary Fig. 4b, c, e, f). This (perhaps expected) observation supports our interpretation that our memory-guided microsaccades were generated voluntarily. Further support for this conclusion is the presence of some express microsaccades after target reappearance (Fig. 2). As stated earlier, because express microsaccades are more likely when a movement plan is already under way[5], the presence of these express microsaccades after target reappearance indicates that the oculomotor system was already planning a corrective movement for any residual errors from previous memory-guided microsaccades. Evidence for such rapid correction also exists in larger saccades (see, for example, ref. [44]).

Naturally, all of the above may be related to questions of awareness. Specifically, and as stated elegantly by Haddad and Steinman[30], labels of involuntary, spontaneous, and reflexive may have arisen in association with microsaccades exactly because people are generally unaware of making these movements. However, we are also very frequently unaware of making larger

saccades. Moreover, these same authors demonstrated[30] that they could be aware of their own self-triggered microsaccades on a surprisingly large fraction of trials. Therefore, awareness on its own is not sufficient to argue that microsaccades are fundamentally different from larger saccades.

Finally, our results pave the way for interesting investigations of spatial working memory at the microscopic scale of the fovea. A wealth of research has been performed on the mechanisms of working memory[36,50] in general, and there were even recent findings that accessing visual working memory engages the oculomotor system[59] (similar to how preparing a manual response in monkeys caused them to generate microsaccades to the four possible previous stimulus locations in ref. [7]). However, just like subcortical mechanisms are less investigated, how foveal space is represented in spatial working memory remains unknown. For example, our perceptual control experiments (Fig. 5) suggest that the overshoot in microsaccade amplitudes that we observed (Fig. 4) was not a deficit in oculomotor control per se. Instead, it likely reflected an expansion of foveal space when working memory was engaged (Fig. 5). This might be a complement of foveal biases in peripheral visual perception[60–62]. Specifically, because of foveal magnification of neural tissue[63], it could be that engaging spatial working memory in the fovea causes distortions in the representation of space, such that foveal locations get mislocalized outward (i.e. overshoot) whereas peripheral locations get mislocalized inward (i.e. undershoot). Interestingly, perceptual mislocalizations around microsaccades[31] are also consistent with outward distortions in the fovea and inward distortions in peripheral vision. This might suggest that there may be generalized underlying mechanisms of perceptual distortions of different kinds (whether due to eye movements, working memory, attention, or otherwise), likely being caused by how individual brain circuits represent visual space in anatomical tissue, and how multiple such circuits (with potentially different levels and kinds of representational magnification) may interact with each other[64].

## Methods

**Ethics approvals**. All monkey experiments were approved by ethics committees at the Regierungspräsidium Tübingen. The experiments were in line with European Union directives, and German laws, governing animal research.

Human experiments were approved by ethics committees at the Medical Faculty of Tuebingen University. Subjects provided written, informed consent, and they were financially compensated for their participation. Our experiments were in accordance with the Declaration of Helsinki.

**Laboratory setups**. Monkey experiments were performed in the same laboratory as that described recently[54,55,65]. Human experiments were done in the laboratory described in refs. [31,64].

Eye movements were recorded at 1 kHz using magnetic induction[66,67] (for monkeys N and M) or video-based eye tracking (for monkey P and the human subjects). Head position in the human subjects was stabilized using a custom head-holder[31].

**Animal preparation**. We collected behavioral data from three adult, male rhesus macaques (macaca mulatta). Monkeys N and M (aged 10 and 7 years, respectively, and weighing 11.5 and 8 kg, respectively) were implanted with scleral search coils to allow measuring eye movements using the magnetic induction technique[66,67]. The eye movements of Monkey P (aged 10 years and weighing 8.6 kg) were recorded using a video-based eye tracker (EyeLink 1000, SR Research). All three monkeys were implanted with a head holder to stabilize head position during experiments, with details on all implant surgeries provided earlier[65,68].

We also recorded SC neural activity from monkeys N and M. The SC was approached through recording chambers cranially-implanted on the midline and aimed at a point 1 mm posterior of and 15 mm above the inter-aural line. The chambers were tilted backwards from vertical by an angle of 35° in monkey N and 38° in monkey M[32,34,55].

**Monkey behavioral tasks**. Monkeys performed a memory-guided saccade task[36,37]. In each trial, a central white fixation spot (72 cd per m$^2$ for monkey N; 86 cd per m$^2$ for monkey M; 48.1 cd per m$^2$ for monkey P) was presented over a

uniform gray background (21 cd per m$^2$ for monkey N; 29.7 cd per m$^2$ for monkey M; 4.4 cd per m$^2$ for monkey P). The fixation spot was a square of 5.3 × 5.3 min arc dimensions. After 300–1,200 ms, a brief target flash (identical to the fixation spot) occurred at some location on the display (duration: ~58 ms), while the fixation spot remained on; this flash represented the location to be memorized by the monkeys for an upcoming saccadic eye movement. After the flash, the fixation spot remained visible for 300–1100 ms; this period constituted the memory interval of the task, after which the fixation spot disappeared. Fixation spot disappearance was the abstract instruction for the monkeys to generate a spatially accurate saccadic eye movement to the remembered flash location. Because of the disappearance of the fixation spot and the flash before it, the saccade that was triggered was generated without any visual guidance. Monkeys had to trigger this saccade within <700 ms from fixation spot disappearance (reaction times were actually much shorter as shown by the distributions in Fig. 2, Supplementary Fig. 2). After gaze entered within a virtual window around the true target location by 300 ms, the target reappeared for 400–500 ms. During this time, a visually-guided saccade, which we also referred to as a corrective saccade, was sometimes triggered to correct for any remaining error between the memory-guided saccade and the actual target position. This corrective movement happened readily, and we did not specifically instruct the monkeys to generate it. It was, however, useful for analysis because individual trials often had both a memory movement as well as a control visually-guided one. The radius of the virtual window around the invisible target position was <3° and was varied depending on target eccentricity. The monkeys were rewarded with water or apple juice at the end of every trial. If they broke fixation during the fixation interval or failed to remain within the virtual window around target location (after the instruction to make the memory-guided saccade), the trial was aborted and repeated.

Across trials, we varied target location (i.e. flash location) between 6 min arc and 16° throughout the entire display. During SC recordings, we made sure to include within our sampling of target locations positions that were in and around the response fields (RF's) of neurons around our recording electrodes, as assessed online using our neurophysiological data acquisition system (see below). We collected a total of 6428 trials in this task from monkey N, 9631 from monkey M, and 14,666 from monkey P.

In monkeys N and M, we also compared memory-guided saccades to those obtained in a task with explicit visual guidance. In this delayed, visually-guided saccade task, stimulus events were identical to those described above, except that the target flash was now replaced by target persistence until trial end. In other words, the target remained on during the memory interval (now called the delay interval) and also after fixation spot disappearance. Thus, the instructed saccade (i.e. after the offset of the fixation spot) was visually-guided. We collected 6147 trials in this task from monkey N and 5871 trials from monkey M.

**Human behavioral tasks**. Human subjects performed the same memory-guided saccade task described above. Target locations were chosen, across trials, from a total of 480 possible pre-defined display positions ranging in eccentricity from 6 min arc to 12°. 288 of these 480 possible target positions were within a rectangle of ±0.8° × ±0.8° size in order to allow us to explore memory-guided microsaccades in particular detail; the remaining locations were equally spaced in the rest of the display. Seven subjects (four naïve and three untrained authors) performed this experiment, with each subject completing 480–600 trials per session and 4–5 sessions (total of 16,844 trials overall). Ten practice trials (with target eccentricities >1°) were performed at the beginning of every session. This task allowed us to relate the monkey behavioral and neurophysiological results to human performance. The subjects were aged 23–39, and three were female.

We also ran two additional control experiments in the humans, in order to understand whether intricacies of memory-guided microsaccade behavior (e.g. Fig. 4) were specific to a particular motor modality (eye movements) or more related to the engagement of spatial working memory per se. We therefore required a perceptual assessment of remembered flash location without the use of an instructed eye movement. In the first control task, which we refer to as the button-press task, trials were identical to the memory-guided saccade task described above up until the end of the memory interval. At this point, instead of fixation spot disappearance, the fixation spot was replaced with a cross-shaped cursor of 59 min arc dimensions. Subjects used a response box with four buttons allowing them to move the cursor at individual pixel resolution (1.46 min arc per click) rightward, leftward, upward, or downward. Individual button presses moved the cursor by individual pixels; prolonged button presses moved the cursor at one of two faster speeds depending on the duration of the press. Once subjects were satisfied with the cursor position as reflecting, as accurately as possible, the remembered flash location, they pressed a fifth button confirming their choice. The target then reappeared at its true position with the cursor disappearing simultaneously. Subjects were free to move their eyes at will when the cursor was visible, and they were given up to 20 s to move the cursor and decide on their perceptual localization of the remembered target location. Therefore, this task allowed assessing the quality of target recall independent of making an instructed memory-guided saccade. Seven subjects (ages: 25–39 years; one female) participated in this experiment, four of which had performed the original saccade version of the task. Each subject participated in 5–6 sessions (250–350 trials per session), for an overall total of 11,249 trials across subjects.

The second control experiment, called the mouse pointer task, was similar to the button-press task except that the cursor was replaced with a computer mouse pointer, which appeared at a random position 1°–5° away from the true flashed target location in every trial. Critically, in this version of the control task, we maintained the fixation spot on during the manual response interval (i.e. until subjects pressed a mouse button), so that an explicit foveal reference frame was always present when subjects were pointing to the remembered target location (similar to how eye movements presumably relied on the initial reference frame of fixation for localization). Subjects were instructed to fixate the fixation spot throughout the response interval until they pressed a mouse button, and they were given up to 15 s to make their response. This task tested whether an explicit foveal reference frame altered the overall pattern of behavior from the button-press task or from the original memory-guided saccade task (Fig. 5). Seven subjects (ages: 25–39 years; one female) performed this task, four of which had also participated in the button-press and memory-guided saccade versions of it. We collected 380–500 trials per session for each subject. We ran five sessions per subject, with the exception of one participant who completed only four sessions (for a total of 16,213 trials overall).

**Monkey neurophysiological recordings.** We recorded SC activity from monkeys N and M using multi-electrode linear arrays (16-channel V-Probes with 150 μm inter-electrode spacing, Plexon). We performed 25 experiments in monkey N and 16 experiments in monkey M. In each experiment, we advanced either one or two V-probes toward SC surface. We monitored the deepest electrode contact in a probe and compared physiological landmarks picked up on this contact to anatomical landmarks in MRI's obtained from the same animals before implant surgeries. Once the deepest electrode contact was in the SC, we advanced the probes further, such that as many electrode contacts as possible were recording SC activity. We used online spike sorting to guide our approach and maximize the number of channels that were recording SC activity. When we inserted two simultaneous V-Probes (23/25 sessions in monkey N), one probe was in the right rostral SC (rostral being the region where microsaccade-related activity is expected[18,19]) and the other was at a slightly more caudal position in the left SC (to allow sampling multiple neuronal preferred eccentricities from the SC's topographic map). When we inserted a single probe, it could be in the right or left SC and at a variety of rostral and slightly more caudal sites.

Once the probes were in the SC, we ran a series of experimental tasks, including the memory-guided and delayed, visually-guided saccade tasks described above. We also ran a variety of other tasks that were not directly relevant to the present study.

Our neurophysiology system sampled electrical activity in each contact at 40 kHz. Spike times during online sorting were sampled at 1 kHz.

**Monkey and human behavioral analyses.** We detected saccades and microsaccades using established methods in our laboratory[68,69]. We manually inspected all trials to correct for false alarms or misses by the automatic algorithms. We also marked blinks or noise artifacts (which were more likely in video-based eye tracking data) for later removal. We analyzed all eye movements occurring in the interval between fixation onset and trial end in the memory-guided and delayed, visually-guided saccade tasks. In the button-press control task, subjects were free to move their eyes after cross-shaped cursor appearance, so we only analyzed eye movements up to such appearance. Individual subjects had idiosyncratic patterns of cursor trajectories and accompanying eye movements that did not appear related in systematic ways to task performance; we thus elected not to analyze eye movements during the response interval in this task. In the mouse pointer task, we monitored eye movements after cursor appearance to ensure that subjects maintained fixation while responding.

In this paper, we report each individual monkey's behavioral performance separately. For the human data, we combined results from all subjects into aggregate measurements. We felt justified in doing so because the human effects exhibited very strong similarities to the monkey behavior, which was a primary reason for performing the human experiments in the first place. Moreover, trial numbers were uniformly distributed across the individual human subjects as much as possible; therefore, no single subject biased the pooled results.

In all tasks, we first identified valid trials that we accepted for subsequent analysis. Valid trials did not have any saccades larger than 1° during the memory or delay interval (to ensure that fixation was maintained properly), and also no blinks or noise artifacts within ±100 ms from target flash time (or target onset time in the delayed, visually-guided version of the task). This latter criterion ensured that subjects could see the target flash well in the memory version of the tasks. We also removed trials if there were blinks or noise artifacts from −100 to +400 ms from the instruction to generate a memory- or visually guided saccade. Because microsaccades alter visibility and modulate visual sensitivity in general[3,33,35,70], we wanted to ensure that subjects could detect target flash (or onset) and fixation spot disappearance (or other instruction to respond) with high sensitivity. We therefore required that valid trials also did not have any microsaccades (or large saccades) within ±50 ms from flash (target) onset or fixation spot disappearance (or any other instruction to respond). Finally, memory-guided saccades had to land within 2° of the target; this constraint was irrelevant for memory-guided microsaccades

(which were much smaller), but it helped ensure that the eye movements were accurate when large target eccentricities were used.

Based on the detected saccades and microsaccades, we computed measures of saccade amplitude, latency (reaction time), direction, starting point, and endpoint. We calculated the starting point as the average eye position in the interval 30 ms before the start of the saccade. The end point was simply the very last sample of the detected saccade. To obtain saccade direction error (i.e. saccade direction relative to target direction from initial fixation spot position), we computed the counter-clockwise angular difference between saccade and target direction. For corrective saccades after target reappearance in the memory-guided saccade task, the target reappeared at a position that was variable relative to gaze position (depending on the landing error of the previous memory-guided movement). We therefore redefined the position of the reappearing target relative to corrective saccade starting eye position. We then calculated saccade direction error as above. We used a similar procedure for target eccentricity. Note that after large memory-guided saccades, target position error for the corrective movements was often small (i.e. large memory-guided saccades were relatively accurate as expected). We therefore had significantly more data for corrective, visually-guided microsaccades than for memory-guided microsaccades (e.g. see the numbers of trials in the legend of Fig. 2).

In the memory-guided saccade task, we defined as the memory-guided saccade (or memory-guided microsaccade) of interest as the first saccade occurring with a latency of at least 50 ms after fixation spot disappearance and less than 50 ms after target reappearance. Similarly, the corrective, visually guided saccade was defined as the first saccade that was executed with latency >50 ms after target reappearance. Because most studies define microsaccades as smaller than 1° in amplitude and we were interested in comparing these to voluntarily generated movements, we treated trials with target eccentricities less than 1° as memory-guided microsaccade trials.

The above analyses were also applied in the delayed, visually-guided saccade task. For the control tasks, we treated button and mouse press locations similarly to how we analyzed saccade endpoints.

In all analyses, we defined trials with target eccentricities <1° as the trials involving microsaccades, as stated above. Corrective, visually-guided saccades were also classified as microsaccades when target eccentricity (relative to eye position at target reappearance) was <1°. As stated above, we used this threshold because it was a typical threshold used in the literature[18,38], but our results (also from other studies) confirm that there is indeed a continuum of saccade generation across different amplitudes.

**Spike sorting for neurophysiological data analyses.** For all neurophysiological data analyses, we performed offline sorting to isolate individual single units. We isolated single units from wide-band (40 kHz) electrode data by adapting the Klusta toolbox[71]. As an additional input to the toolbox, we recreated the electrode geometry (16 channels with linear spacing) such that neighboring channels could share spike timing and waveform information (as in, virtual tetrodes). From filtered neural data (750–5000 Hz), the toolbox performed automated spike detection and classification. We inspected the resulting cluster assignments (putative units) using the Kwik-gui interface included in the Klusta toolbox. Based on auto- and cross-correlograms, as well as waveform shapes, we manually curated each cluster in order to obtain well-isolated units. Clusters that showed no characteristic auto-correlogram, or that produced atypical spike waveforms, were clustered together, labeled as noise, and subsequently excluded from further analysis. We estimated single unit isolation quality of all resulting units using established metrics[72,73]. Specifically, we computed an estimate of the false positive rate based on refractory period violations (inter-spike interval violations). Furthermore, we obtained an isolation distance measure for each cluster, by calculating the Mahalanobis distance between any two units. All isolated units that exceeded an estimated false positive rate of 10% or had an isolation distance below 30 were excluded from further analysis; all other units are hereafter termed putative neurons (297 neurons in total; we selected for analysis all 188 neurons that showed clear SC visual and/or saccade-related RF's encompassing microsaccadic amplitude ranges, as assessed by the procedures described below). Spike trains of all putative neurons were then convolved with an appropriately-scaled Gaussian ($\sigma = 10$ ms) to obtain an estimation of the neuron's firing rate in spikes per s.

**Neurophysiological data analyses.** We classified SC neurons as visual, visual-motor, or motor depending on visual and saccade-related responses. We defined several 100-ms intervals based on key trial events. The visual response interval comprised the time from 50 to 150 ms after target onset. The motor interval was set to be 100 ms centered around saccade onset. We also measured this saccade-related interval for corrective saccades in the memory-guided saccade task. Since we were especially interested in characterizing microsaccade-related discharge, we classified movement-related aspects of neurons based on responses associated with saccades <1° in amplitude. To estimate baseline activity during fixation without any target or saccade onset, we used the 100-ms steady fixation interval before target onset. We avoided contamination of baseline and visual neural activity by saccades or microsaccades; we excluded all trials in which such movements occurred during the measurement intervals. Similarly, we excluded trials in which there was more than one movement within the motor interval. In all cases, we measured average firing rate within a given measurement interval, which resulted in a population of

measurements across trials; we only included neurons with >10 repetitions per measurement interval. We compared each of visual, memory-guided microsaccade, corrective microsaccade, or visually-guided microsaccade (in the delayed version of the task) activity to baseline activity using one-tailed Wilcoxon-ranksum tests ($\alpha = 0.05$). We then grouped neurons into five categories: (1) Visual (V-Neuron), possessing only a visual increase relative to baseline; (2) Visual-motor (VM-Neuron), possessing a visual increase and either memory- or corrective-(visual) saccade response; (3) Motor (M-Neuron), exhibiting both memory- and corrective-(visual) saccade response but no visual response; (4) Visually dependent (VDSR-Neuron), possessing a saccade-related response only when a visual target was present for the eye movement[46–48]; (5) Exclusively-responding (ER-Neuron), exhibiting a microsaccade-related response only in the memory-guided microsaccade task but not for similarly sized visually-guided microsaccades.

RF hotspot locations (e.g. Fig. 7f) were estimated based on all of the sampled saccade vector amplitudes (and corresponding flash locations for visual responses) in the memory-guided saccade task (even > 1°); our estimates may therefore include neurons that may potentially have responded even more strongly had we obtained even larger eccentricities (at the very least, all neurons in Fig. 7f were related in one form or another to microsaccades). For every isolated neuron that had a motor response, we also estimated the saccadic amplitudes that caused the neuron to discharge, also known as the neuron's movement field. For every neuron, we compared the population of baseline values as described above to the trial-by-trial responses for saccades. We then calculated the movement field by a linear-interpolation of all z-transformed saccade-related responses. Locations with z-scores > 1.96 were considered to be part of the movement field. We plotted movement fields using log-polar coordinates, to magnify small eccentricities associated with microsaccades, as described previously[19]. Movement-related hotspots were identified based on the stronger of either the memory-guided or the visually guided movement-related responses.

**Statistical analyses**. We performed statistical tests (behavioral and neural) with an $\alpha$ level of 0.05 unless otherwise noted. When the assumptions of normality were not met for a two-sampled t-test, we used a Wilcoxon ranksum test instead. For tests related to saccade directions, we used the CircStat toolbox[74]. When comparing proportions (for example, for success rate), we used a $\chi^2$-test.

**Literature meta-analysis**. We searched for and read articles published between 1965 and November 2017. We used the Web of Science search database, and we employed the keyword "microsaccade*". We then selected, out of 467 research articles in the global results, a total of 342 articles that were genuinely related to microsaccades and that were also written in English. Any article that referred to microsaccades as being "voluntary" (20/31 articles), "controlled" (6/31 articles), or "suppressed voluntarily" (5/31 articles) was classified in the "voluntary" category (31 articles). Even though "suppressed voluntary" is conceptually different from the voluntary control that we were interested in for the current study, we nonetheless included this description in the voluntary category to give this category the benefit of the doubt in our meta-analysis when comparing to "involuntary" descriptions. In contrast to 31 articles, there were 148 articles in the "involuntary" category (almost 5 times as many). For the "involuntary" category (148 articles), we included any article using the following descriptors for microsaccades: "involuntary" (134/148 articles), "spontaneous" (7/148 articles), "unaware" (3/148 articles), "automatic" (2/148 articles), "unconscious" (1/148 articles), "stochastic" (1/148 articles).

**Reporting summary**. Further information on research design is available in the Nature Research Reporting Summary linked to this article.

## Data availability

All data presented in this paper are stored in institute computers and are available upon reasonable request.

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

## Acknowledgements
We were funded by the Werner Reichardt Centre for Integrative Neuroscience (CIN). The CIN is an Excellence Cluster (EXC307) funded by the Deutsche Forschungsgemeinschaft (DFG). We were also supported by the Hertie Institute for Clinical Brain Research at Tuebingen University, and DFG-funded Research Unit (FOR1847; project: HA6749/2–1).

## Author contributions
X.T., A.B., Z.M.H. collected neurophysiological data. X.T., J.B., Z.M.H. collected monkey behavioral data. K.F.W., A.R-C. collected human behavioral data. K.F.W., A.B., Z.M.H. analyzed data. Z.M.H. wrote paper. K.F.W., X.T., A.B., A.R-C., Z.M.H. edited paper.

## Additional information

**Competing interests:** The authors declare no competing interests.

