## [Peer Review File · Nature Communications]

Editorial Note: Parts of this Peer Review File have been redacted as indicated as we could not obtain permission to publish the reports of reviewer 2. An additional reviewer was added in the second round of review.

Reviewers' Comments:

Reviewer #1:

Remarks to the Author:

This manuscript presents psychophysical and neural data showing that, contrary to conventional, "well established" wisdom, microsaccades (saccadic eye movements of < 1 degree of amplitude) are voluntarily controlled, very much like standard, large saccades. The behavioral data, based on the classic memory-guided saccade task, are convincing, abundant (monkeys and humans), and include important control conditions (corrective saccades, visually guided microsaccades). The neural data make an important contribution too, by revealing new response types in the SC that are specific to memory-guided or visually-guided microsaccades. Overall this is a strong package, the sort of study that will make many people think, "how come I didn't think of this?" It should definitely change the way we think about microsaccades.

I only have a few minor comments.

(1) Early on, it would be important to stress that saccade targets were presented at random locations, and that macro- and micro-saccade trials were interleaved. This is in the Methods, but it's a key aspect of the task that should be highlighted.

Related to this, for the 3 left columns of Fig. 3, it wasn't clear to me whether 0 degrees was the actual target location or whether the data were rotated to align all the targets at zero. It's fine either way, but should be specified.

(2) Another aspect of the task that should also be highlighted in the main text is the criterion for considering a response successful (second and third columns of Fig. 2). This must involve a spatial and a temporal window, but in particular the spatial criterion probably required some care in the case of microsaccades. I wasn't sure what that was (or might have missed it).

(3) It seems that the idea that microsaccades are involuntary is one of those that was established based on scant evidence and then took a life of its own; i.e., it got repeated over and over with little questioning of the original evidence. To be fair, though, even in the case of macro-saccades, the majority of them are made without much thinking, and in some cases we may be unaware that they occur. As an example, Klapetek et al. (2016) observed that, after an error, participants were often unaware of making corrective saccades. What I wonder is whether our threshold for awareness is higher for microsaccades than for large saccades. For instance, if I fixate continuously on the o within the word "pop" (at reading distance), I'm not sure if it is just my attention that wanders a bit toward the p's or whether my eyes move -- a sort of ambiguity that is much less likely when stimuli are further away from the fixation point. This is just to say that although microsaccades are voluntary, perhaps they are still substantially different from larger saccades in some ways (e.g., awareness). The consistent overshoot found for the smallest amplitudes already hints at this. If they have any thoughts or data, the authors may want to comment on this in the Discussion, to balance it a bit (relative to lines 561-564).

(4) The two blue colors in Supplementary Figure 3 are very similar (at least on my printed hardcopy). It would be nice if the color difference could be enhanced.

Emilio Salinas

Reviewer #2:

Remarks to the Author:

[Redacted]

**Responses to Reviewer Comments on:
“Memory-guided microsaccades”
by Willeke*, Tian*, Buonocore*, Bellet, Ramirez-Cardenas, and Hafed
April 3, 2019**

We thank both reviewers very much for their insightful comments, which we believe have improved our manuscript. We have now fully addressed all of the reviewer comments. In what follows, we provide specific responses to the reviewer comments, and we also highlight the line numbers for the actual edits made in the revised manuscript. We sincerely hope that the manuscript is now ready for publication, and we thank the reviewers and editors one more time for the encouragement and support to improve the quality of the manuscript.

Responses to Reviewer #1

REVIEWER COMMENT: This manuscript presents psychophysical and neural data showing that, contrary to conventional, "well established" wisdom, microsaccades (saccadic eye movements of < 1 degree of amplitude) are voluntarily controlled, very much like standard, large saccades. The behavioral data, based on the classic memory-guided saccade task, are convincing, abundant (monkeys and humans), and include important control conditions (corrective saccades, visually guided microsaccades). The neural data make an important contribution too, by revealing new response types in the SC that are specific to memory-guided or visually-guided microsaccades. Overall this is a strong package, the sort of study that will make many people think, "how come I didn't think of this?" It should definitely change the way we think about microsaccades.

OUR RESPONSE: Thank you very much for this sentiment. We really do appreciate it very much, and we also appreciate the perspective of Reviewer 2, who has raised important points that we agree with (because they have actually been the exact source of motivation for our study). As you will hopefully see from the revised manuscript and the responses below, we have now fully addressed all of your and Reviewer 2's comments.

REVIEWER COMMENT: I only have a few minor comments.

(1) Early on, it would be important to stress that saccade targets were presented at random locations, and that macro- and micro-saccade trials were interleaved. This is in the Methods, but it's a key aspect of the task that should be highlighted.

OUR RESPONSE: We have now added this information at the beginning of the Results section. The text now reads: "After initial training with large saccades, we tested the monkeys on pseudorandomly chosen stimulus locations, including at small eccentricities typically associated with microsaccades. We specifically interleaved small and large eccentricities across trials within any given session,

and we also sampled different visual field locations (in the left/right or upper/lower visual fields). We did so in order to avoid overtraining any specific target locations or eccentricities.” Please see lines 97-102 of the revised manuscript. We have also kept the original mention in the Methods section, as you stated.

REVIEWER COMMENT: Related to this, for the 3 left columns of Fig. 3, it wasn't clear to me whether 0 degrees was the actual target location or whether the data were rotated to align all the targets at zero. It's fine either way, but should be specified.

OUR RESPONSE: We apologize for any potential source of confusion here. The data were plotted as a “direction error” measure. We now see that the figure legend of Fig. 3 may have been even more clear on this than in our original manuscript (even though the figure itself labels the angular measurement unit as “direction error” and the associated main text describes it as “angular directional difference between memory-guided microsaccades and target locations”). Therefore, in the revised manuscript, we have now added more information to the figure legend. Please see lines 1195-1197 of the revised manuscript. In the main text, we also added an additional clarifying statement. Please see lines 180-183 of the revised manuscript.

REVIEWER COMMENT: (2) Another aspect of the task that should also be highlighted in the main text is the criterion for considering a response successful (second and third columns of Fig. 2). This must involve a spatial and a temporal window, but in particular the spatial criterion probably required some care in the case of microsaccades. I wasn't sure what that was (or might have missed it).

OUR RESPONSE: Once again, apologies for any lack of clarity. We have now modified the text in several places to avoid any confusion by our readers. Specifically, we have modified the legend of Fig. 2 (please see line 1178 of the revised manuscript). We have also edited the main text around Fig. 2 to clarify our definition of “success rate” for this figure (please see lines 141-142, 150-152 of the revised manuscript). And, we have modified the main text around Fig. 3 to relate success rate in Fig. 2 to directional accuracy (please see lines 180, 200-202 of the revised manuscript).

To briefly answer your question here, for Fig. 2, we defined “success” solely based on reaction time. That is, was there a movement “induced” by the instruction within a reasonable reaction time? In later figures, we found that these movements (i.e. the ones that passed the reaction time criterion) were also overwhelmingly spatially accurate. We felt that this is the fairest way of demonstrating the spatial accuracy of the movements. If we had introduced a spatial criterion for success in Fig. 2, then subsequent results on spatial accuracy (e.g. Fig. 3) would be somewhat trivial and less convincing.

REVIEWER COMMENT: (3) It seems that the idea that microsaccades are involuntary is one of those that was established based on scant evidence and then took a life of its own; i.e., it got repeated over and over with little questioning of the original evidence. To be fair, though, even in the case of macro-saccades, the majority of them are made without much thinking, and in some cases we may be unaware that they occur. As an example, Klapetek et al. (2016) observed that, after an error, participants were often unaware of making corrective saccades. What I wonder is whether our threshold for awareness is higher for microsaccades than for large saccades. For instance, if I fixate continuously on the o within the word "pop" (at reading distance), I'm not sure if it is just my attention that wanders a bit toward the p's or whether my eyes move -- a sort of ambiguity that is much less likely when stimuli are further away from the fixation point. This is just to say that although microsaccades are voluntary, perhaps they are still substantially different from larger saccades in some ways (e.g., awareness). The consistent overshoot found for the smallest amplitudes already hints at this. If they have any thoughts or data, the authors may want to comment on this in the Discussion, to balance it a bit (relative to lines 561-564).

OUR RESPONSE: We think that the more defining/important factors in scenarios like the ones that you have mentioned in your example might have to do more with things like spatial resolution than with a fundamental difference between small and large saccades. For example, if you introspect about where you are looking in the two scenarios that you mention in your example (fixating the "o" in a single word "pop", versus fixating and introspecting about where something is that is much farther away from fixation), then spatial resolution likely gives a very strong hint about where you are looking. In the latter case, the much farther word or object in the periphery would not be as discernible as the letters "p" in the word "pop", and this is enough to know that you are not looking directly at the peripheral stimulus.

In fact, in the original Steinman work, which we had cited in our original manuscript (Haddad and Steinman, 1973), they made an explicit point that subjects could indeed be aware of making their own microsaccades. So, it seems to us that awareness is another, almost orthogonal, dimension from voluntary control. Moreover, this other dimension (of awareness) is related to small and large saccades in similar ways, supporting our overall view.

Similarly, the overshoot that we have observed likely has to do with foveal magnification in neural tissue, which we find to be much larger in the superior colliculus than previously believed (Chen et al., BioRxiv, 2019), and not necessarily with a categorical difference between small and large eye movements.

Having said all of the above, we have now added a full paragraph in Discussion, exactly in line with your advice. Please see lines 521-532 of the revised manuscript.

REVIEWER COMMENT: (4) The two blue colors in Supplementary Figure 3 are very similar (at least on my printed hardcopy). It would be nice if the color difference could be enhanced.

OUR RESPONSE: Thank you. We agree, and we have now updated the figure with a new color (please see the new Supplementary Figure 3).

Responses to Reviewer #2

[Redacted]

Reviewers' Comments:

Reviewer #1:

Remarks to the Author:

With their revisions, the authors addressed all the comments I had made. I think this is a strong manuscript that will be interesting for both its behavioral and neural data. It presents strong converging evidence for the idea that the difference between microsaccades and standard 'large' saccades is just their size.

Emilio Salinas

Reviewer #3:

Remarks to the Author:

The authors seek to show that microsaccades are comparable to saccades in the degree to which they are subject to volitional control.

To demonstrate "at will" initiation of microsaccades, monkeys and humans performed a "memory-guided" saccade task (saccading to a remembered target location at eccentricities ranging from 6' to 5° of arc) as well as several control tasks (visually-guided saccades, memory-guided manual movements). For all eccentricities, saccade reaction times were comparable, although saccade probability declined somewhat with eccentricity. Moreover, saccade reaction times were comparable to corrective saccades and to visually-guided saccades (when express saccades were disregarded). Directional accuracy of memory-guided saccades were significantly better than that of (uninstructed) corrective saccades and comparable to (instructed) visually-guided saccades. Importantly, directional accuracy was comparable in all directions (not just cardinal directions). The amplitude of memory-guided microsaccades scaled with target eccentricity (albeit with a systematic bias towards larger amplitudes at the small end of the range). Memory-guided button presses and mouse pointing (by human observers) showed a very similar dependence, suggesting a systematic mis-representation of small eccentricities in working memory ("foveal expansion"). In contrast, the amplitude of uninstructed, corrective saccades was unrelated to target eccentricity at the small end of the range. These results show convincingly that saccades of all sizes (down to 6' of arc) may readily be initiated "at will" in a timely and accurate fashion (in terms of both direction & amplitude).

In addition to this behavioural evidence from monkeys and humans, the authors performed neurophysiological recordings from superior colliculus in monkeys. SC neuron activity accompanied memory-guided saccades of all sizes (to targets within a neuron's receptive field). Interestingly, some neurons (ER-neurons) fired with (instructed) memory-guided saccades, but not with (non-instructed) visually-guided corrective saccades, whereas other neurons (VDSR-neurons) showed the opposite association. Comparison of peri-stimulus or peri-microsaccadic activity also showed other neurons firing only in association with visual stimulation (V-neurons), or with oculomotor response (M-neurons), or with both (VM-neurons). This demonstrates, in addition to previously known SC neuron types (V-, M-, VM-neurons) the existence of a previously unknown cohort of SC neurons associated specifically with memory-guided saccades (22% ER-neurons) as well as the existence of neurons associated specifically with small visually-guided saccades (30% VDSR-neurons). Intriguingly, the authors suggest that the engagement of different SC populations in different task settings (memory-guided, visually-guided, etc) may reflect different representations of visual space in CORTICAL populations mediating each task.

In conclusion, the manuscript presents compelling evidence that saccades of all amplitudes (including microsaccades) may be performed volitionally, dispelling the persistent misperception of microsaccades as being involuntary. This is an important and timely contribution, especially in view of the current resurgence of this field. The manuscript also presents behavioural evidence (systematic overshoot) that different tasks do not rely on identical foveal representations (such as memory- and visually-guided saccades). Importantly, the manuscript presents corroborating neurophysiological evidence that memory-guided and visually-guided microsaccades engage overlapping (though not identical) collicular populations. Thus, the manuscript avoids the simplistic message of "saccades of all amplitudes are generated by the same mechanisms" and instead offers the far more interesting

message of "... generated by similar mechanisms involving task-specific foveal representations". The methodological thoroughness of the paper is exemplary and its nuanced and differentiated message most welcome in this age of oversimplification!

Remarks:

Line 311 & 312: The remark pertaining to memory-guided manual tasks is placed oddly and should perhaps be moved to Line 305.

Jochen Braun

**Responses to Reviewer Comments on:
“Memory-guided microsaccades”
by Willeke*, Tian*, Buonocore*, Bellet, Ramirez-Cardenas, and Hafed
July 12, 2019**

Responses to Reviewer #1

REVIEWER COMMENT: With their revisions, the authors addressed all the comments I had made. I think this is a strong manuscript that will be interesting for both its behavioral and neural data. It presents strong converging evidence for the idea that the difference between microsaccades and standard 'large' saccades is just their size.

OUR RESPONSE: Thank you very much for this sentiment.

Responses to Reviewer #3

REVIEWER COMMENT: The authors seek to show that microsaccades are comparable to saccades in the degree to which they are subject to volitional control.

To demonstrate “at will” initiation of microsaccades, monkeys and humans performed a “memory-guided” saccade task (saccading to a remembered target location at eccentricities ranging from 6' to 5° of arc) as well as several control tasks (visually-guided saccades, memory-guided manual movements). For all eccentricities, saccade reaction times were comparable, although saccade probability declined somewhat with eccentricity. Moreover, saccade reaction times were comparable to corrective saccades and to visually-guided saccades (when express saccades were disregarded). Directional accuracy of memory-guided saccades were significantly better than that of (uninstructed) corrective saccades and comparable to (instructed) visually-guided saccades. Importantly, directional accuracy was comparable in all directions (not just cardinal directions). The amplitude of memory-guided microsaccades scaled with target eccentricity (albeit with a systematic bias towards larger amplitudes at the small end of the range). Memory-guided button presses and mouse pointing (by human observers) showed a very similar dependence, suggesting a systematic misrepresentation of small eccentricities in working memory (“foveal expansion”). In contrast, the amplitude of uninstructed, corrective saccades was unrelated to target eccentricity at the small end of the range. These results show convincingly that saccades of all sizes (down to 6' of arc) may readily be initiated “at will” in a timely and accurate fashion (in terms of both direction & amplitude).

In addition to this behavioural evidence from monkeys and humans, the authors performed neurophysiological recordings from superior colliculus in monkeys. SC neuron activity accompanied memory-guided saccades of all sizes (to targets

within a neuron's receptive field). Interestingly, some neurons (ER-neurons) fired with (instructed) memory-guided saccades, but not with (non-instructed) visually-guided corrective saccades, whereas other neurons (VDSR-neurons) showed the opposite association. Comparison of peri-stimulus or peri-microsaccadic activity also showed other neurons firing only in association with visual stimulation (V-neurons), or with oculomotor response (M-neurons), or with both (VM-neurons). This demonstrates, in addition to previously known SC neuron types (V-, M-, VM-neurons) the existence of a previously unknown cohort of SC neurons associated specifically with memory-guided saccades (22% ER-neurons) as well as the existence of neurons associated specifically with small visually-guided saccades (30% VDSR-neurons). Intriguingly, the authors suggest that the engagement of different SC populations in different task settings (memory-guided, visually-guided, etc) may reflect different representations of visual space in CORTICAL populations mediating each task.

In conclusion, the manuscript presents compelling evidence that saccades of all amplitudes (including microsaccades) may be performed volitionally, dispelling the persistent misperception of microsaccades as being involuntary. This is an important and timely contribution, especially in view of the current resurgence of this field. The manuscript also presents behavioural evidence (systematic overshoot) that different tasks do not rely on identical foveal representations (such as memory- and visually-guided saccades). Importantly, the manuscript presents corroborating neurophysiological evidence that memory-guided and visually-guided microsaccades engage overlapping (though not identical) collicular populations. Thus, the manuscript avoids the simplistic message of "saccades of all amplitudes are generated by the same mechanisms" and instead offers the far more interesting message of "... generated by similar mechanisms involving task-specific foveal representations".

The methodological thoroughness of the paper is exemplary and its nuanced and differentiated message most welcome in this age of oversimplification!

Remarks:

Line 311 & 312: The remark pertaining to memory-guided manual tasks is placed oddly and should perhaps be moved to Line 305.

OUR RESPONSE: Thank you! We really do appreciate this support and encouragement very much.

Concerning the sentence in lines 311-312: We have now followed your advice and moved this remark to a point earlier in the paragraph, exactly as suggested. The new location of the sentence in the revised manuscript (in the Word file with "track changes" turned on) is now lines 388-389.